# Root Foraging Strategy Improves the Adaptability of Tea Plants (*Camellia sinensis* L.) to Soil Potassium Heterogeneity

**DOI:** 10.3390/ijms23158585

**Published:** 2022-08-02

**Authors:** Li Ruan, Hao Cheng, Uwe Ludewig, Jianwu Li, Scott X. Chang

**Affiliations:** 1Key Laboratory of Soil Contamination Bioremediation of Zhejiang Province, Zhejiang A&F University, Hangzhou 311300, China; ruanl@zaas.ac.cn; 2Institute of Sericulture and Tea, Zhejiang Academy of Agricultural Sciences, Hangzhou 310021, China; 3National Center for Tea Improvement, Tea Research Institute, Chinese Academy of Agricultural Sciences, Hangzhou 310008, China; chenghao@tricaas.com; 4Institute of Crop Science, Nutritional Crop Physiology, University of Hohenheim, Fruwirthstr. 20, 70593 Stuttgart, Germany; u.ludewig@uni-hohenheim.de; 5Department of Renewable Resources, University of Alberta, Edmonton, AB T6G 2E3, Canada

**Keywords:** cellulose decomposition, F1 hybrid population, low-K tolerant, organic acids, root development

## Abstract

Root foraging enables plants to obtain more soil nutrients in a constantly changing nutrient environment. Little is known about the adaptation mechanism of adventitious roots of plants dominated by asexual reproduction (such as tea plants) to soil potassium heterogeneity. We investigated root foraging strategies for K by two tea plants (low-K tolerant genotype “1511” and low-K intolerant genotype “1601”) using a multi-layer split-root system. Root exudates, root architecture and transcriptional responses to K heterogeneity were analyzed by HPLC, WinRHIZO and RNA-seq. With the higher leaf K concentrations and K biological utilization indexes, “1511” acclimated to K heterogeneity better than “1601”. For “1511”, maximum total root length and fine root length proportion appeared on the K-enriched side; the solubilization of soil K reached the maximum on the low-K side, which was consistent with the amount of organic acids released through root exudation. The cellulose decomposition genes that were abundant on the K-enriched side may have promoted root proliferation for “1511”. This did not happen in “1601”. The low-K tolerant tea genotype “1511” was better at acclimating to K heterogeneity, which was due to a smart root foraging strategy: more roots (especially fine roots) were developed in the K-enriched side; more organic acids were secreted in the low-K side to activate soil K and the root proliferation in the K-enriched side might be due to cellulose decomposition. The present research provides a practical basis for a better understanding of the adaptation strategies of clonal woody plants to soil nutrient availability.

## 1. Introduction

Tea is, after water, the world’s most popular beverage. It contains abundant secondary metabolites such as polyphenols, theanine, and caffeine, which offer a wealth of human health benefits. In the last decade, worldwide tea production has increased by ~66% in acreage and has reached 5.3 million tons on 3.5 million hectares across 50 tea-growing countries (Food and Agriculture Organization of the United Nations statistics. https://www.fao.org/faostat (accessed on 6 July 2022)). Potassium (K) is one of the most important nutrients for the growth of tea plants. The availability of K in the soil greatly affects the quality, yield and resistance to pests and abiotic stresses of tea plants [1,2]. Soils in tea plantations are generally highly acidic and have high leaching rates, resulting in the loss of soil K. In addition, a large amount of K is exported from the ecosystem by tea leaf harvesting. The percentage of tea gardens that are deficient in soil K increased from 59% in the 1990s to 74% in the 2010s in China [3]. Due to weathering, atmospheric deposition, leaching, fertilization, tillage and other processes, K availability in most managed soils is spatially heterogeneous [4,5]. For example, no-tillage combined with strip fertilization, and the accumulation of litter caused by artificial pruning can both increase the K heterogeneity [6,7]. Tea plantations are usually distributed in mountainous or hilly areas with steep slopes. Those tea plantations are usually managed without tillage but with strip fertilization and litter accumulation are common, factors that often increase the spatial heterogeneity of K in the soil.

The plasticity of plant roots in response to the fluctuating environmental conditions is one of the main mechanisms by which plants optimize their water and nutrient acquisition [4]. When the soil K is heterogeneous, roots will adopt a series of plastic foraging strategies to increase the absorption of K [8,9]. The plasticity to foraging mainly includes the following strategies: (1) root morphological remodeling, that is, increasing root length and surface area in K-enriched soil patches and limiting root growth in the K-deficient patches [10,11]; (2) release of root exudates to improve the availability of soil K, so that different forms of soil K can be used by plants [12,13]; and (3) gene expressions of calcium signaling, respiration, proton pumps, K transporters and channels to sense external K availabilities and respond to their changes at the molecular level [14,15,16]. Differences in root traits, such as root diameter among different plant species, affect the absorption of K and foraging strategies of roots to soil nutrient heterogeneity [17,18]. 

Previous studies on root foraging strategies for K were mainly focused on annual plants, such as *Arabidopsis*, *Glycine max*, *Triticum aestivum*, *Zea mays* and *Oryza sativa* [19,20]. There are few studies on tea plants, which are perennial and are mainly reproduced by asexual propagation. Adventitious roots of asexually propagated tea seedlings grow laterally, and their geotropism is weak. Therefore, the adaptation of tea roots to soil K heterogeneity may be very different to other crops. Furthermore, previous studies were mainly focused either on the roles of root morphology, root exudation, or genes for improving plant acclimation to soil K heterogeneity, but whether there is coordination among these processes remains to be verified.

In this paper, we studied root foraging strategies of tea plants under soil K heterogeneity. Two tea genotypes from the F1 hybrid population of Longjing 43 × Baihaozao hybrid were selected for this research. The two genotypes had a similar genetic background but were very different in their tolerance to low soil K availability and root development, including a low-K tolerant genotype “1511” and a low-K intolerant genotype “1601” [16,21]. Tea plants are highly heterozygous because of their incompatibility and frequent interspecific hybridization. Taking this specific breeding line as the research object makes this study based on a similar genetic background, which is more conducive to finding out the target genes. As reported in previous studies, the reason for the strong adaptabilities to changes in nutrient availabilities of plants with different tolerances is that their roots have strong plasticity, so that they can uptake more nutrients from regions with high nutrient availability [22,23]. For example, *C. quinoa*, as a plant that is highly tolerant to low K availability, responds to K heterogeneity through strong root plasticity [24]. *Pinus massoniana,* with a different tolerance to low nutrient availability, also responds differently to nutrient heterogeneity by various root foraging strategies [25]. Thus, when the soil K availability is heterogeneous, plants with different tolerances are likely to respond to K heterogeneity through root morphological and physiological plasticity. 

The starting hypothesis is that the tea genotype “1511” that is tolerant to low K availability adjusts better to the heterogeneity of soil K by adopting a root foraging strategy that optimally coordinates root morphology and physiology. The differences in root foraging strategies between the two tea genotypes were compared by monitoring root morphologies, organic acid contents in root exudates, the soil K solubilizing abilities of root exudates, and the contents of different K fractions in soils over a 12-month period in a greenhouse incubation experiment using a multi-layer split-root system (Figure 1). The key genes regulating foraging strategies were identified by transcriptome sequencing. Our findings will not only contribute to our further understanding of the adaptation strategies of tea plants to soil K availability, but also help to provide a reference for tea growers and producers to improve the potassium utilization by making full use of tea roots. 

## 2. Results

### 2.1. The Nutrition and Growth of Tea Plants in the Multi-Layer Split-Root System

After one year of growth in the multi-layer split-root system, there were no significant differences in leaf, stem and root K concentrations between the two tea genotypes in the U.K+ treatment (Figure 2A). The leaf K concentrations were 47 and 38% higher in “1511” than those of “1601” in the Sp.K+/Sp.K0 and U.K0 treatments, respectively (*p* < 0.01). The leaf K concentrations in the Sp.K+/Sp.K0 treatment were the same as those in the U.K+ treatment for “1511”, but was only 67% in the former than in the latter for “1601”. The root K concentrations were lower in “1511” than in “1601” in the Sp.K+ treatment (*p* < 0.01), but higher in “1511” than in “1601” in the U.K0 treatment (*p* < 0.01) (Figure 2A). The K biological utilization indexes (KUIB) were higher in “1511” than in “1601” in the Sp.K+/Sp.K0 and U.K0 treatments (Figure 2B, *p* < 0.05), but not different between the two genotypes in the U.K+ treatment. Moreover, in both shoot and root, the activities of major antioxidant enzymes (SOD, POD and CAT) were significantly higher in “1511” than those in “1601” when both were stressed by K limitation. For the shoot system, the content of procyanidins in “1511” was significantly higher than that in “1601” under the K limitation, while there was no significant difference in total phenol content between the two tea plant types. For the root system, the contents of both procyanidins and total phenol in “1511” were significantly higher than those in “1601” under K limitation. Therefore, the two genotypes had different tolerances for the K limitation (Appendix A).

### 2.2. Temporal and Spatial Variations of Soil K Concentrations and Root Development

The slowly available K was higher at the end than at the beginning of the experiment in all treatments (Figure 3), but the temporal pattern for available K was the opposite (Figure 4). The slowly available K and available K changed as a function of the distance from roots. Under homogeneous K supply (U.K+ and U.K0), the slowly available K concentration depends on the distance from the roots of “1511”, but under heterogeneous K supply (Sp.K+/Sp.K0) the spatial variation was low around the roots of “1511”. On the contrary, under heterogeneous K supply (Sp.K+/Sp.K0), the above changes in “1601” were not reduced (Figure 3).

The total root length reached the maximum in February for both genotypes. However, the total root length was the longest in the Sp.K+ treatment for “1511” (Figure 5A), but not in the U.K+ treatment for “1601” (Figure 5B). Total root length was not different between Sp.K0 and U.K0 throughout the study for “1511”, but not different between Sp.K+ and Sp.K0 for “1601”. For “1511”, the total root length was higher (*p* < 0.05) in Sp.K+ than in Sp.K0 in February, May, and November. The fine root (<1 mm) length was higher in Sp.K+ than in U.K+ throughout the study for “1511” (Figure 5C), but not for “1601” (Figure 5D).

### 2.3. Release of Organic Acid and Solubilization of Soil K by Root Exudates

Oxalic and formic acids were the main organic acids secreted by the roots of the two genotypes (Appendix A). The total organic acid content reached the maximum in August for both genotypes (Figure 6A). For “1511”, the order of total organic acid content was Sp.K0 > Sp.K+ > U.K+ > U.K0. For “1601”, the order of total organic acid content was Sp.K+ > U.K0 > Sp.K0 > U.K+. The amount of soil K solubilized by root exudates was similar to the amount of organic acid secreted across the treatments (Figure 6B). The solubilization of soil K was the highest in the Sp.K0 treatment for “1511”, but not in the Sp.K+ treatment for “1601”.

### 2.4. Potential Key Genes Involved in Root K Foraging 

There were more low-K responsive genes in “1511” than in “1601”. Moreover, there were more low-K responsive genes involved in regulating the responses to K heterogeneity in “1511” than in “1601”. For “1511”, starch and sucrose metabolism, phenylpropanoid biosynthesis, and flavonoid biosynthesis were the most significant enriched KEGGs. For “1601”, there were no significant enriched KEGGs. The enrichment of starch and sucrose metabolism was found in both tea genotypes, while the enrichment degree and the number of “1511” were higher than those of “1601” (Figure 7).

The relationships and expression patterns of the key genes enriched in the starch and sucrose metabolism pathway are shown in Figure 8. Some of these genes (TEA026325, TEA008079, TEA19040) were responsible for converting cellulose to D-glucose, and other genes (TEA014414, TEA023084) were responsible for converting starch and glycogen to ADP-glucose. The expression levels of all these genes were higher in K heterogeneous than in K homogeneous conditions only for “1511”. Genes (TEA026325, TEA008079, TEA19040) that are responsible for converting cellulose into D-glucose also had higher expression levels in “1511” than in “1601” under heterogeneous K conditions (Figure 8).

Through genome annotation and differential gene analysis, a total of 1445 candidate genes were screened, which were linked to the K tolerance. Among these genes, there were 252 (17.44%) newly discovered genes. Based on the comparison of reference genome, 1185 SNP loci were found in the candidate genes (82.01%). Among the 1185 SNP loci, 46.24% had an impact on the expression product, of which the variation of nine loci (0.76%) had a high impact on the expression product, which will be focused on in the future. The relevant information of the potential molecular markers of K tolerance for tea plants is shown in Appendix A.

## 3. Discussion

Plant root systems can sense changes in the availability of soil nutrients and then adapt to these changes through root morphological plasticity [26]. Previous studies have found that root plasticity, which is generally manifested by increasing root growth in K-enriched patches and decreasing root growth in K-deficient patches, plays an important role in adapting to heterogeneity in K availability in various plants [10,27,28]. In this study, the low-K tolerant genotype “1511” effectively increased the root growth on the K-enriched side throughout the year, but the low-K intolerant genotype “1601” lacked this ability; our data thus add to the literature that the morphological plasticity is strongly related to the tolerance of plants to the nutrient in short supply. The expression of root morphology depends on the plant nutritional status and perception of local nutrients, so as to make better use of the nutrients with heterogeneous distributions in the soils [4]. The abilities of roots to locate nutrients depends on plant genotypes [19]. The distribution of more roots on the K-enriched side is conducive to the efficient acquisition of K resources for “1511”. There is a higher total root length of “1511” in Sp.K+ than in U.K+, while the total root length of “1511” in Sp.K0 did not change significantly as compared with U.K0. Roots can sense local nutrient changes and integrate these local signals into the system signal system [29]. The heterogeneous K distribution promoted the root developments on the K-enriched side in “1511”, suggesting that the root morphological plasticity of “1511” was likely caused by the active foraging behavior of roots on the K-enriched side that was stimulated by the low-K signal from the Sp.K0 side. 

Compared with nitrogen and phosphorus, potassium has unique characteristics in plants. Potassium mainly exists in an ionic state and does not form organic compounds in plants. Therefore, compared with the major nutrient nitrogen and phosphorus, potassium is relatively active and easy to flow, and can be rapidly redistributed in plants [30]. Under such circumstances, it is possible for the K absorbed by roots in the Sp.K+ side to be transported to roots in the Sp.K0 side, which resulted in a higher K concentration in roots on the Sp.K0 side than that on the U.K0 side in both genotypes. Root growth needs material consumption and results in better nutrient absorption. The whole plant system weighs the relationship between material loss and nutrient uptake. This trade-off ability determines its adaptability to the nutrient environment to a certain extent [20]. In February, May and November, the low-K tolerant genotype “1511” preferred to promote the root growth in the Sp.K+ side rather than in the Sp.K0 side. However, there was no significant difference in the root growth between the Sp.K+ and Sp.K0 side for the low-K intolerant tea genotype “1601”. This suggested that the low-K tolerant genotype “1511” might better balance the relationship between the material consumption and nutrient absorption. In addition, the carbon consumption is relatively low during the fine root formation, which helps to maintain high K absorption efficiency [31,32]. The fine root (<1 mm) length was higher in Sp.K+ than in U.K+ throughout the study for “1511”, but this did not happen in “1601”. Those might be important reasons for the high K efficiency of the low-K tolerant genotype “1511” under spatial K heterogeneity.

Through the release of root exudates to increase the solubility of nutrients, plants effectively adapt to the poor soil nutrition environment [33,34]. For example, low-K signal can promote the secretions of organic acids by roots and improve K availability in soils [33,34]. This study found that the secretion of organic acids from the roots of tea plants helped to improve soil K availability. There were great differences between the secretion amounts of organic acids from roots induced by homogeneous (U.K0) and heterogeneous (Sp.K0) low-K signals. Low-K signals stimulate roots to secrete organic acids, which consume a lot of carbon and energy [35]. K is required for carbon and energy consumption [36]. The Sp.K0 side had both low-K signal and K supplements as material energy support from the Sp.K+ side. Therefore, in this study, the heterogeneous low-K signal (Sp.K0) could stimulate roots to secrete organic acids more than the homogeneous low-K signal (U.K0), especially for the low-K tolerant genotype. In addition, the low-K signal stimulates roots to secrete organic acids in order to improve the effectiveness of K. It will cause a waste of material and energy when large amounts of organic acids are secreted in the K-rich patches [37]. In this study, root organic acid secretion in the low-K tolerant genotype was more targeted than in the low-K intolerant genotype under K heterogeneity, resulting in more organic acids secreted in the Sp.K0 side rather than in the Sp.K+ side. 

In the low-K tolerant genotype, a series of genes involved in the decomposition of cellulose to glucose were highly expressed under K heterogeneity, especially on the Sp.NK side, with expression levels higher than that of the low-K intolerant genotype. The degradation of root cellulose makes the root cell wall soft, which is conducive to the rearrangement and growth of root cells [38,39,40]. The K heterogeneity induced the expression of cellulose decomposition genes in the roots of “1511” on the Sp.NK side. This might be an important reason for promoting the root growth of “1511” on this side. In addition, glucose is one of the main sugar sources in plants; glucose can promote root development [41,42]. For “1511”, a series of genes that are involved in the decomposition of cellulose to glucose and the transformation of starch and glycogen to ADP-glucose were highly expressed, as induced by K heterogeneity; those genes likely have promoted the growth of the root of “1511”.

Previous studies on other plants showed that the root K foraging strategies mainly included morphological remodeling, physiological regulation and gene regulation, which were coordinated with each other [20]. There are obvious differences in coordination among different plants. Some studies believe that the above differences mainly depend on the root diameters [23]. In this study, the low-K tolerant tea genotype developed a perfect coordination mechanism (Figure 9), which included developing more roots on the K-enriched side (Sp.K+) to improve K acquisition, secrete more organic acids by roots to increase soil K solubility on the K-deficient side (Sp.K0), and the cellulose decomposition genes expressed on the K-enriched side to promote the root proliferation. In order to confirm this statement, we conducted a real time PCR on three genes involved in cellulose decomposition, and found that the results of the real time PCR were significantly positively correlated with the total root length (Appendix A). The coordination among root K foraging strategies for tea plants was similar to that for *Arabidopsis*, which relied on root morphological remodeling and physiological regulation at the same time [10,43]. However, the gene regulations of tea plants focused on the root morphological remodeling. The coordination of tea plant was very different from those of *Triticum aestivum*, *Zea mays* and *Glycine max*, which mainly relied on one K foraging strategy (root morphological remodeling, physiological regulation or gene regulation) [19,20]. This seemed to be related to the coexistence of fine and thick roots in tea plants.

K transporters and channels are critical in the K^+^ absorption and translocation in plants [30]. In our study, a K transport related gene (TEA016242) was significantly up-regulated in “1511”, while there was no K transport related gene up-regulated in “1601” under K stress. Meanwhile, the leaf and root K concentrations were significantly higher in “1511” than those in “1601”. Therefore, for K absorption, the molecular and physiological results were consistent under K stress. When plants are stressed by K deficiency, ROS will be produced in large quantities. SOD is the first defense line to remove ROS in plant cells [44]. In this study, an SOD gene (TEA008360) was significantly up-regulated only in “1511” under K stress. For “1601”, no genes related to antioxidation were found under K stress. Meanwhile, the activity of SOD was significantly higher in “1511” than that in “1601” when both roots were stressed by K limitation. Therefore, for the antioxidation process, the molecular and physiological results were also consistent under K stress.

This study revealed how tea plants improve their absorption of K under heterogenous K supply through a root foraging strategy. We concluded the low-K tolerant tea genotype acclimated to soil K heterogeneity through its root foraging strategies, including remodeling of root morphology (more fine roots) and effective soil K solubilization by root exudates. The high expression of genes that decomposed cellulose to glucose in roots may be an important mechanism for promoting root proliferation on the K-enriched side. Although the root system of the clonal tea plant has poor geotropism, this does not affect their root foraging for soil K. This research provides a better understanding of root strategies to soil nutrient heterogeneity of woody plants with a different tolerance to low K availability. In the absence of any help of soil microorganisms, soil chemistry and climate change will also have significant impacts on the K availability for the tea plants, while this study mainly focused on the role of strategic plants in improving soil K utilization. Therefore, tea farmers should also fully consider the impact of soil chemistry and climate change on soil nutrient availability.

## 4. Methods and Materials

### 4.1. Plant and Soil Materials

Among the 327 individuals of the F1 hybrid population produced from crossing Longjing 43 × Baihaozao, two tea genotypes, “1511” and “1601”, which had very different tolerance levels to low-K availability and root development, were identified. The “1511” was a tea genotype tolerant to low K availability with good root development, while “1601” was intolerant low K availability, with weak root development [16,21]. One-year-old tea cuttings of those two genotypes were used for this research employing multi-layer split-root boxes. The soil used in the multi-layer split-root box was collected from a tea garden in Meijiawu (30°11′26″ N, 120°4′58″ E), Hangzhou, Zhejiang province. The soil type was Orthic Anthrosols according to the Chinese Soil Taxonomy [45]. The soil physical and chemical properties were as follows: pH 4.8, organic matter content 34.7 g kg^−1^, alkaline hydrolysable N 66.4 mg kg^−1^, available K 72.6 mg kg^−1^, available P 69.2 mg kg^−1^, total N 4.7 g kg^−1^, soil texture was clay loam according to the international soil texture classification [46]. The collected soil samples were air-dried, passed through a 2 mm sieve and sterilized by γ-irradiation (>50 kGray) (Xiyue Radiation Technology Co., Ltd., Nanjing, China) before use.

### 4.2. Experiment on Soil Culture with Multi-Layer Split-Root Box

The spatial heterogeneity of K in soil was simulated by a multi-layer split-root box. The root box was 400 mm long, 160 mm wide, and 120 mm high. The root box was divided into two parts along the width side by a PVC plate to allow different K supplies in the chambers to be implemented. The root box was further divided into different microzones 20 mm wide using a 30 μm nylon net. The design of the root box was based on a previous study [47]. The split-root rhizobox is used to study the morphological, physiological and biochemical characteristics of roots. It can be used to study individual environmental factors (such as soil potassium). The plants growing in the split-root rhizobox are easy to master, and the growth conditions can be repeated. However, there are also some limitations, which can raise questions about the validity of the study. Therefore, field experiments will be carried out later to further verify the wide application of this study. In the multi-layer split-root system (Figure 1A), heterogeneous and homogeneous (uniform) K environments were simulated as follows (Figure 1B): (1) U.K+: both compartments had K fertilizer applied, (2) Sp.K+/K0: one compartment had K fertilizer applied, while the other did not, and (3) U.K0: both compartments had no K fertilizer applied. Each treatment included fifteen root boxes, which included three experimental replicates and five destructive sampling times. Twelve one-year-old tea seedlings were grown in each root box. In the U.K+ treatment, 0.745 g K, 1.665 g N, and 0.727 g P, applied as potassium sulphate, urea, and superphosphate, respectively, were added to each root box. In the U.K0 treatment, 1.665 g N and 0.727 g P were applied to each root box. In the Sp.K+/K0 treatment, 0.373 g K, 0.833 g N and 0.364 g P were applied to half of the root box, and 0.833 g N and 0.364 g P were applied to the other half of the root box [48]. At the beginning of the experiment, all fertilizers were mixed into the soil and put into the respective compartments of the root boxes. Each root box was filled with 9 kg of soil. The experiment started on 1 November 2018, and ended on 30 November 2019, conducted in a greenhouse in the Tea Research Institute, Chinese Academy of Agricultural Sciences, in Hangzhou, Zhejiang Province. Although it is a greenhouse, we opened the shelters around when the experiment on soil culture with multi-layer split-root box was carrying out. It was actually a greenhouse with light transmission, rain protection and wind ventilation. We did this to keep the temperature in the greenhouse roughly consistent with the outside world. The greenhouse temperatures during the growth period of the experiment were 12–18 °C (November (begin)), 5–9 °C (February), 17–26 °C (May), 25–33 °C (August) and 10–19 °C (November (end)), respectively.

### 4.3. Determinations of Soil K Contents, Root Morphology, and Plant K Contents

Soil samples were destructively collected five times, on 1 November 2018, 28 February 2019, 31 May 2019, 31 August 2019, and 30 November 2019. At each sampling, soil samples from different microzones were collected separately. For the 1 November 2018 sampling, fertilizers had been mixed into the soil without planting the tea plants. When collecting soil samples from root boxes with tea plants, tea plants were dug carefully from the soil and the soil on the roots was brushed away carefully. In the above process, the roots were not separated from the tea plant. The soil brushed away from the roots was mixed with the soil from the corresponding microzone. All twelve tea plants were collected from each root box at destructive harvesting. The collected soils were air-dried and passed through a 1 mm sieve for determining soil K content. For the determination of soil available K, the soil samples were extracted by 1 mol L^−1^ NH_4_OAc. Then the K concentration in the extracts was determined by a flame photometer (Model 425, Sherwood Scientific Ltd., Cambridge, UK). For the determination of slowly available K, the test soils were extracted by 1 mol L^−1^ hot HNO_3_ (120–130 °C). Then the K concentration in the extract was determined by the flame photometer described above [49]. The difference between the acid soluble K and available K is the slowly available K.

The roots were washed and scanned with a root scanner (EPSON Expression 1640XL, Seiko Epson Corp., Nagano, Japan), and then the scanned root images were analyzed with an image analysis system (WinRHIZO, Regent Instruments Inc., Quebec City, QC, Canada). After that, the root, stem, and leaf samples were dried in an oven at 80 °C for 48 h and were weighed. For the determination of K concentrations in plant samples, the dried plant samples were digested by H_2_SO_4_ and H_2_O_2_, and then the K in digested solutions was measured by the flame photometer described earlier. 

The K biological utilization index was calculated as follows [50]:(1)$$K\;\mathrm{biological}\;\mathrm{utilization}\;\mathrm{index}=\mathrm{shoot}\;\mathrm{dry}\frac{\mathrm{weight}}{\mathrm{shoot}}K\;\mathrm{concentration}.$$

### 4.4. Root Exudate Collection and Soil K Solubilization Test

A micro root exudate collector [51] was embedded in the rhizosphere of the mini root box (Appendix A). The mini root box used for the in-situ collection of root exudates was 100 mm long, 160 mm wide and 120 mm high. An annular hollow fiber sampler (Kuraray super fine filter 301, Kuraray Co., Ltd., Tokyo, Japan) connected with a syringe, which is called a root exudate collector, was preinstalled horizontally in the soil at the middle height of the mini root box (Appendix A). Three root boxes (i.e., three replicates) were set up for each treatment, and each root box contained three tea plants. 

Soil solutions in root boxes without tea plants were also collected and treated as the background root exudation. According to different fertilization treatments, the soil solutions of these blank controls were labeled as blank (U.K+), blank (U.K0), blank (Sp.K+) and blank (Sp.K0). Due to the fertilization treatment differences, there were also differences between the blank soil solutions without tea plants. Thus, four blanks were used in this experiment. On the day before the collection of root exudates, the soil moisture in the root box was adjusted to 80% field capacity. Thus, more root exudates can be dissolved into the soil solution and collected by the annular hollow fiber sampler [51]. On the day the root exudates were collected, a 10 mL soil solution was collected with 100 kPa suction. Collections were conducted in November 2018, February 2019, May 2019, August 2019, and November 2019. We called the above process of adjusting soil moisture one day and collecting root exudates the next day one collection process. The collection process was repeated 15 times in the collection month. The collected soil solutions were freeze-dried and stored in a refrigerator at −20 °C for organic acid determination and a soil activation test. The organic acid content in the soil solution was determined by high-performance liquid chromatography (HPLC) measured at a wavelength of 210 nm [52].

In the soil K activation experiment, 1.0 g freeze-dried root exudates and 5.0 mL distilled water were added into a 2.5 g air-dried soil sample. In addition, the freeze-dried soil solutions collected from root boxes without tea plants were also used in the soil K activation experiment as the blank [53]. At the same time, blank control (CK, using distilled water) was also performed. The air-dried soils from the root boxes without receiving fertilizers were used in the soil K activation experiment. All treatments were replicated three times. The soil K activation experiment was carried out in a growth chamber, with the temperature set to 25 °C and the relative humidity at 70%. The water content in those soils used for the activation experiment was adjusted to 80% field capacity every day. After fifteen days of activation, the soil samples were collected and extracted by 1 mol L^−1^ NH_4_OAc. Then the available K concentration in the extracts was determined by the flame photometer described above [3]. The soil K activation ability of root exudates was evaluated by the soil available K content after the soil K activation experiment [54]:(2)$$\mathrm{Soil}\;K\;\mathrm{activation}\;\mathrm{ability}\;={\;\mathrm{Available}\;K}_{Root\;exudate}-{\mathrm{Available}\;K}_{Blank}.$$

### 4.5. Culture Experiment with a Split-Root System

Because there were many interference factors in the soils, in order to better clarify the molecular adaptation mechanism of tea roots to K heterogeneity, a nutrient solution culture experiment with relatively few interference factors was performed as follows: one-year old tea cuttings were cultivated with the roots evenly planted on the two sides of the split root hydroponic box with a root canal. Three treatments were included to simulate the heterogeneous and homogeneous K environments as follows: (1) a homogeneous K condition (U.NK: both compartments had a K nutrient solution added); (2) a heterogeneous K condition (Sp.NK/Sp.SK: one compartment had a K nutrient solution added, while the other had a K starvation nutrient solution added); and (3) a homogeneous K starvation condition (U.SK: both compartments had a K starvation nutrient solution added). The K nutrient solution and K starvation nutrient solution contained 1 mM and 0 mM K_2_SO_4_, respectively. The composition of other nutrients in the hydroponic solution was as follows [54]: 1.0 mM NH_4_NO_3_, 0.035 mM CaH_4_O_4_P_2_, 0.67 mM MgSO_4_, 0.495 mM CaCl_2_, 0.035 mM Al_2_ (SO_4_)_3_·18H_2_O, 7.0 × 10^−3^ mM H_3_BO_4_, 1.0 × 10^−3^ mM MnSO_4_·H_2_O, 6.7 × 10^−4^ mM ZnSO_4_·7H_2_O, 1.3 × 10^−4^ mM CuSO_4_·5H_2_O, 4.7 × 10^−5^ mM (NH_4_)_6_Mo_7_O_24_·4H_2_O and 4.2 × 10^−3^ mM EDTA-FeNa. Before the treatments were imposed, the tea plants were first cultured in the K nutrient solution until fresh white roots had well developed; this process took 60 days. During the pre-culture period, the nutrient solutions were changed once a week. Then the three treatments were imposed for 5 d, during which time the nutrient solutions were changed once a day. At the end of the experiment, the roots on the two sides of the split root hydroponic box were collected separately, washed and quickly put into liquid nitrogen. The experiment was completed in a growth chamber in the greenhouse of the Tea Research Institute between 3 February 2019 and 8 April 2019. According to the previous research [55], the temperature of the growth chamber was 25 °C during the day and 22 °C at night. The relative humidity, day/night period and light intensity were 70%, 14/10 h and 200 mmol m^−2^ s^−1^, respectively. The nutrient solution in each split root hydroponic box was continuously ventilated by a pump. Three replicates were established for each treatment, with twelve tea plants included in each root box.

### 4.6. Transcriptome Analysis and qRT-PCR Verification

The collected root samples were ground with mortar in liquid nitrogen. The total RNA was extracted from roots with a TRIZOL reagent (Cat#15596-018, Life Technologies, Carlsbad, CA, USA), following the operating instructions from the manufacturer. The degradation and contamination of RNA were monitored on 1% agarose gel. The purity of RNA was checked by a NanoPhotometer^®^ spectrophotometer (NP80, IMPLEN, Inc., Munich, Germany). The RNA integrity was evaluated using RNA Nano 6000 Assay Kit of Bioanalyzer 2100 system (Agilent Technologies, Palo Alto, CA, USA) [9].

The construction of the cDNA library and the RNA sequencing were performed on the Illumina Hiseq (2500, Illumina, San Diego, CA, USA) following the operation manual. The raw data in fastq format (raw reading) were first processed by an internal Perl script. Specifically, clean data (clean read) were obtained by removing the adapter-containing read, aggregation-containing read and low-quality read from the original data. At the same time, the clean data of Q20, Q30, and GC were obtained. All downstream research used the cleaned high-quality data. The gene model annotation file and reference genome [56] were directly obtained from the genome website (http://tpia.teaplant.org/index.html accessed on 10 November 2021). The reference genome index was established and the paired-end clean reading accepted the alignment process with the reference genome in Histat 2v2.0.5. Genes with P-adj < 0.05 and |log_2_(FoldChange)| > 0 were defined as differentially expressed genes. The sets of the raw data and normalized expression data were deposited in the Gene Expression Omnibus (GSE198198) at the National Center for Biotechnology Information (https://www.ncbi.nlm.nih.gov/geo/query/acc.cgi?acc=GSE198198 accessed on 14 May 2022).

For the enrichment analysis of KEGG (Kyoto Encyclopedia of Genes and Genomes), the cluster analysis tool in R was used. The KEGG refers to a database resource that groups genes into categories according to cellular function and biological mechanisms from molecular level information, especially for large datasets established by genome sequencing and other high-throughput experimental techniques (http://www.genome.jp/kegg/ accessed on 28 November 2021). To better understand the responses of low-K up-regulated genes to K heterogeneity, the Sp.NK vs. U.NK up-regulated genes and Sp.SK vs. U.SK down-regulated genes shared with the low-K up-regulated genes were selected. Meanwhile, to better understand the responses of low-K down-regulated genes to K heterogeneity, the Sp.NK vs. U.NK down-regulated genes and Sp.SK vs. U.SK up-regulated genes shared with the low-K down-regulated genes were selected. The KEGG enrichment analysis was carried out for the above selected genes. Because the K signals from the two sides of the roots will affect each other, the KEGG comparison of the above genes can effectively identify the potential key genes for K heterogeneity.

To study the reliability and repeatability of transcriptome data, we randomly selected several genes from the two tea genotypes for real-time fluorescence quantitative polymerase chain reaction. The remaining RNA samples in the transcriptome analysis were used for RT-PCR analysis to ensure the reliability and repeatability of the results. The PCR primers were designed using the Primer 5 and DNAMAN software (Appendix A). According to the manufacturer’s instructions, in a reaction volume of 20 μL, a rapid quantitative reverse transcription kit (Tiangen, China) was used to synthesize the first strand gene, with a total of 800 ng of RNA as a template. The ABi7500 RT-PCR machine based on the SYBR Green reagent was used for a rapid qPCR test (Takara, Japan) in 20 μL reactions. The *GAPDH* was used as a reference gene for standardizing the expression of target genes [57]. The experiment was performed with three technical repetitions and three experimental replicates. To analyze the quantitative changes of genes, the relative differences were analyzed with the quantitative 2^−ΔΔCt^ method. RNA-seq data had a linear relationship with qRT–PCR (r^2^ = 0.76) (Appendix A), which confirmed the reliability of the RNA-seq data.

### 4.7. Determinations of Antioxidant Enzyme Activities and Polyphenol Contents

The determination methods have also been added in the Methods as follows: in order to analyze the physiological responses of the two tea plant types when both were stressed by K limitation, a culture experiment was carried out as described by our previous study [16]. First, tea plants were grown in normal nutrition solution for four weeks. Second, half of the above tea plants were transferred to a nutrient solution without K_2_SO_4_ (i.e., K starvation treatment) (SK), while the other half was transferred to a nutrient solution with K_2_SO_4_ as a control (i.e., control treatment) (CK). The tea plants were grown in normal and SK for eight weeks and the shoot and root were collected separately to measure the antioxidant enzyme activities and polyphenol contents. The activities of SOD, POD and CAT were determined according to previous methods [58,59]. The procyanidin and total phenol contents were analyzed by the methods described by the previous research [60]. 

### 4.8. Statistical Analysis

For statistical analysis, Microsoft Excel (Microsoft Corporation, Redmond, WA) and SPSS Windows 18 edition (SPSS Incorporation, Chicago, IL, USA) were employed. A one-way analysis of variation (ANOVA) was used to analyze the treatment effects on the K biological utilization index, total root length, and solubilization of soil K by root exudates (available K). An independent sample *t*-test was performed to test the differences of the leaf, stem and root K concentrations between the two tea genotypes. Before the analyses of one-way ANOVA and independent sample *t*-test, the normality of distribution and homogeneity of variance were tested. Among all the data, only the data of stem K concentration did not meet the above assumptions. Therefore, the reciprocal of sine was used to transform the data, and then the difference of the transformed data was analyzed. The original data rather than the transformed data were reported in this paper. The analysis of the transcriptome data was carried out with the help of the Novomagic Analysis System (https://magic.novogene.com accessed on 17 November 2021). Origin 9.0 (Origin Inc., Chicago, IL, USA) was used for drawing the figures.

## Figures and Tables

**Figure 1 ijms-23-08585-f001:**
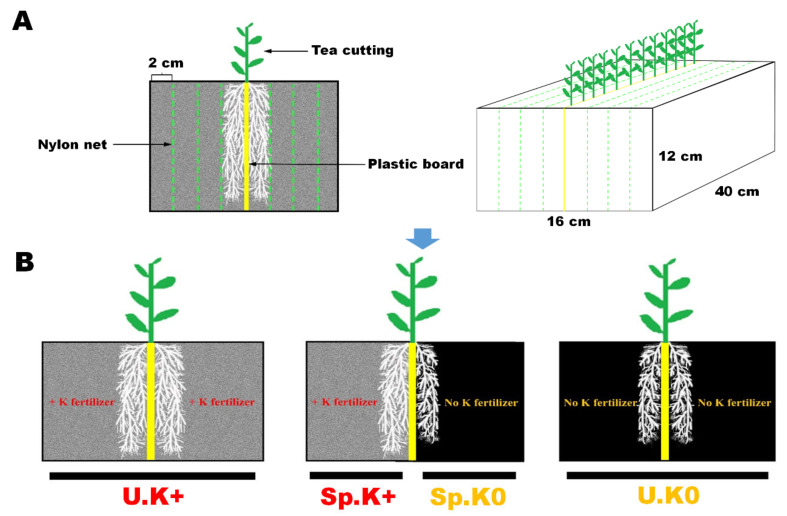
Multi-layer split-root box and treatments. Multi-layer split-root system (**A**). Treatments of heterogeneous and homogeneous K environments (**B**).

**Figure 2 ijms-23-08585-f002:**
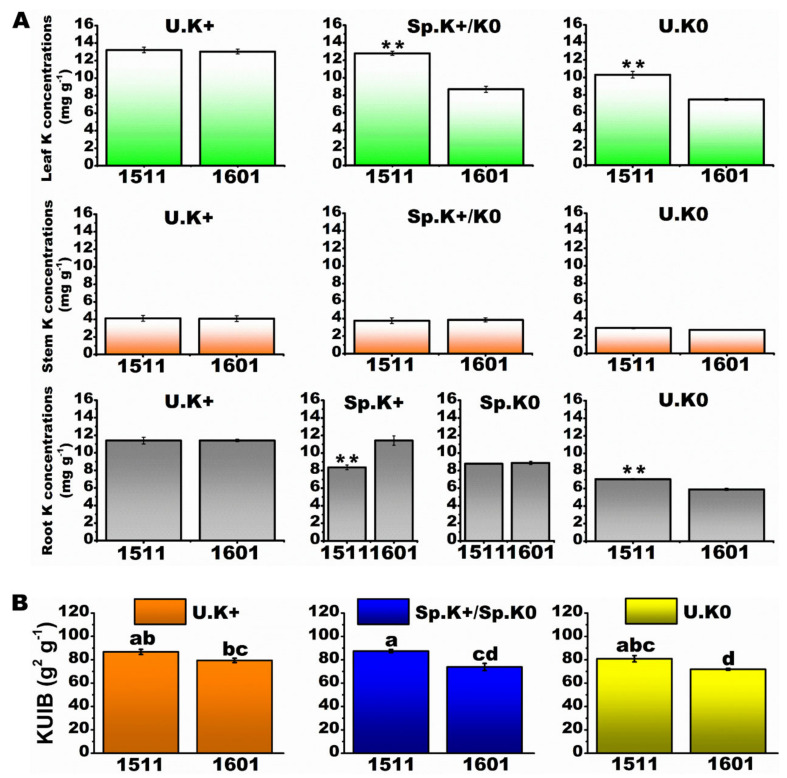
Growth of tea plants under heterogeneous and homogeneous K environments. K concentrations in different tissues under different treatments (**A**). K biological utilization indexes of tea plants under different treatments (**B**). Data are means ± SE (*n* = 3). Different letters represent significant differences at the level of *p* < 0.05. ** represents a highly significant difference between the two genotypes under *p* < 0.01.

**Figure 3 ijms-23-08585-f003:**
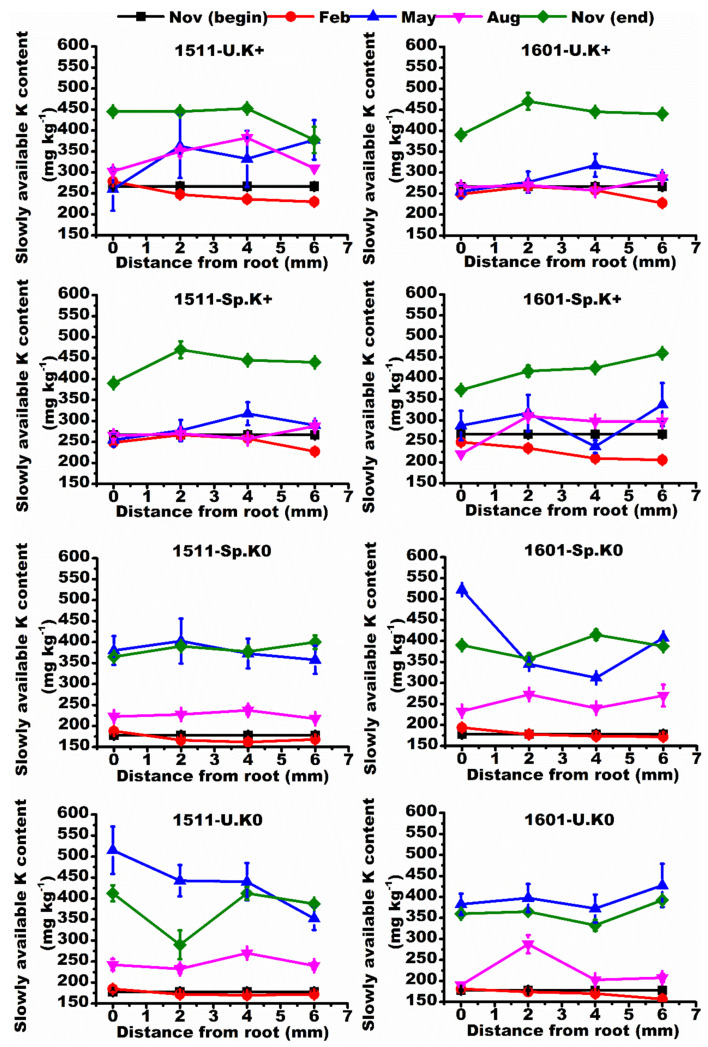
Spatiotemporal variations of slowly available K contents in soils. Different colored lines represent the spatial distribution of soil slowly available K in different months. Data are means ± SE (*n* = 3).

**Figure 4 ijms-23-08585-f004:**
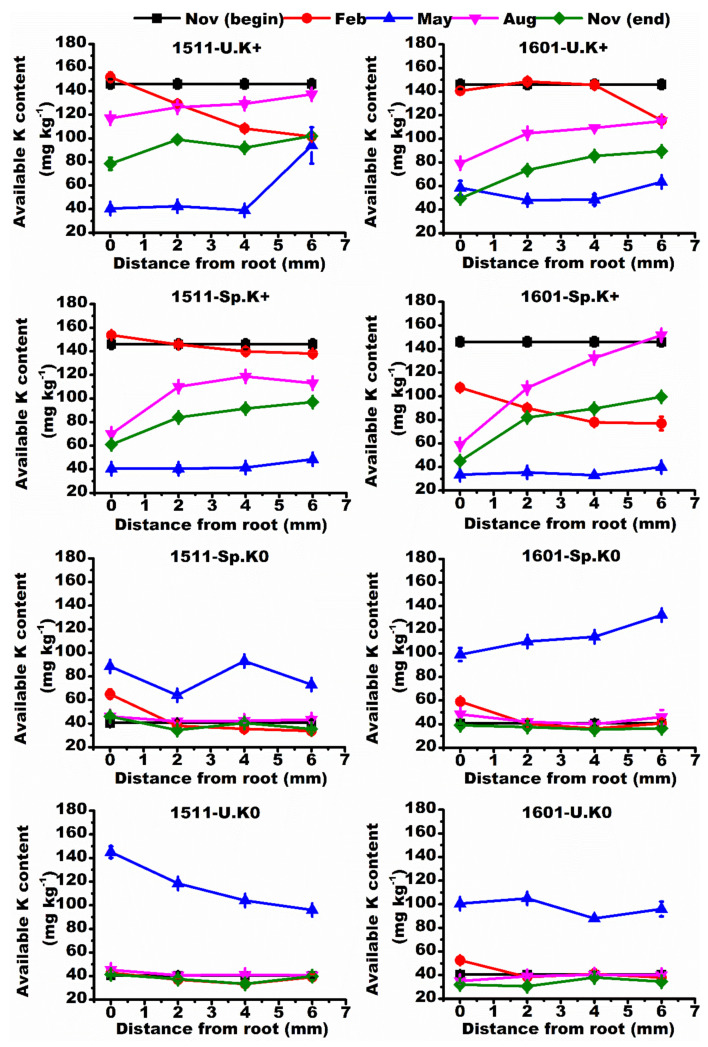
Spatiotemporal variations of available K contents in soils. Different colored lines represent the spatial distribution of soil slowly available K in different months. Data are means ± SE (*n* = 3).

**Figure 5 ijms-23-08585-f005:**
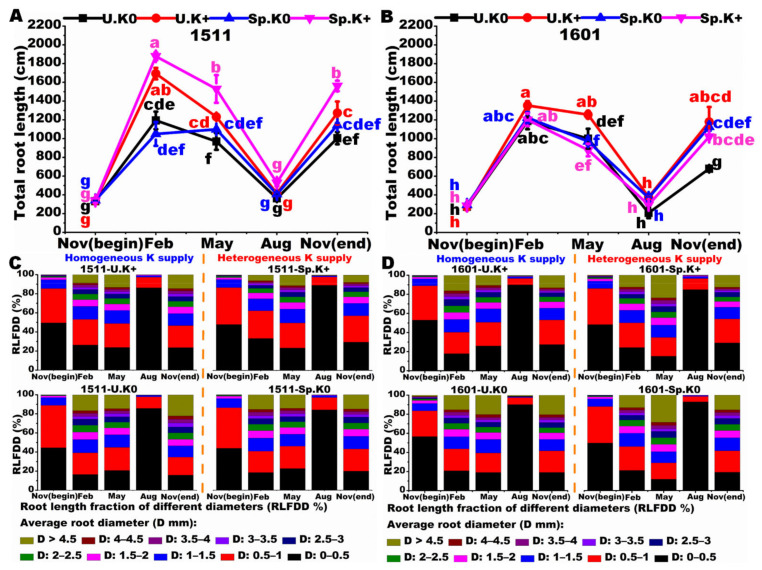
Temporal variations of total root length and root length fractions of tea plants. Temporal variation of total root length of genotype “1511” (**A**). Temporal variation of total root length of genotype “1601” (**B**). Root length fraction changes of each diameter of genotype “1511” (**C**). Root length fraction changes of each diameter of genotype “1601” (**D**). The letter colors were consistent with the representative colors of the treatments. Different letters represent significant differences at the level of *p* < 0.05. Data are means ± SE (*n* = 3).

**Figure 6 ijms-23-08585-f006:**
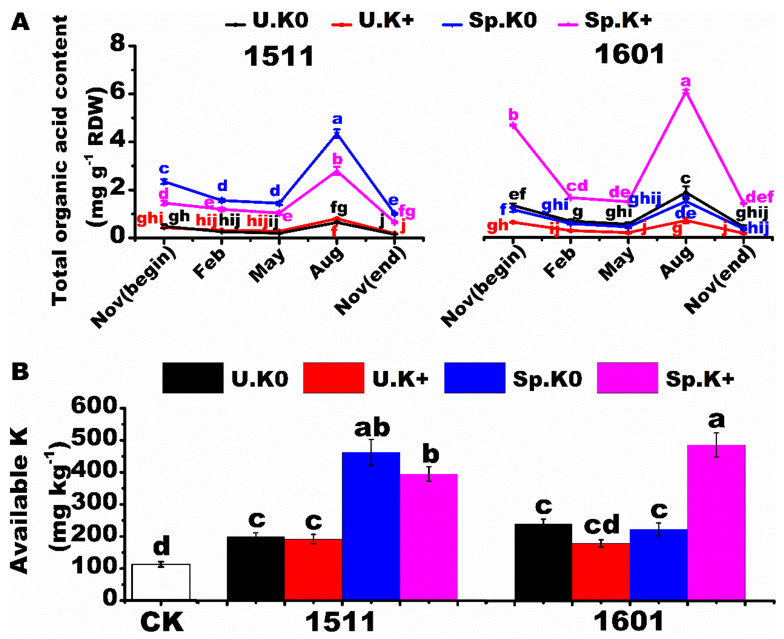
Temporal variations of organic acids in root exudates and solubilization of soil K by root exudates. Temporal variation of total organic acid content of root exudates (**A**). Solubilization of soil K by root exudates (**B**). CK treatment represents the soil without root exudates. U.K0, U.K+, Sp.K0 and Sp.K+ treatments represent the soils with root exudates collected from U.K0, U.K+, Sp.K0 and Sp.K+ treatments in the mini root boxes. The letter colors were consistent with the representative colors of the treatments. Data are means ± SE (*n* = 3). Different letters represent significant differences at the level of *p* < 0.05.

**Figure 7 ijms-23-08585-f007:**
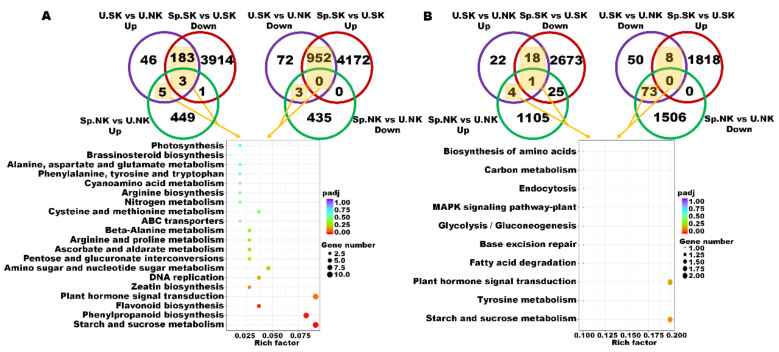
KEGG pathway enrichment analyses of the two tea plant genotypes in response to heterogeneous vs. homogeneous K environments. KEGG pathway enrichment analyses in genotype “1511” to acclimate to heterogeneous K supply (**A**). KEGG pathway enrichment analyses in genotype “1601” to acclimate to heterogeneous K supply. The key genes for KEGG pathway enrichment analyses were circled in the Venn diagrams (**B**).

**Figure 8 ijms-23-08585-f008:**
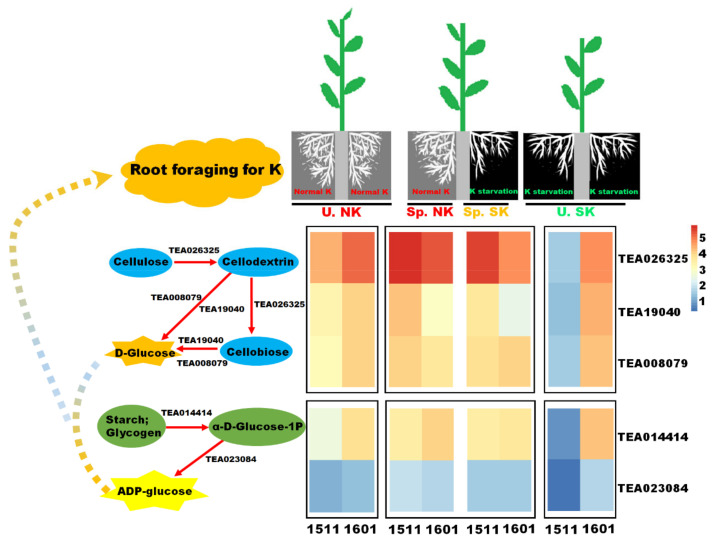
Relationships and expression patterns of the key genes enriched in the starch and sucrose metabolism pathway in heterogeneous vs. homogeneous K environments.

**Figure 9 ijms-23-08585-f009:**
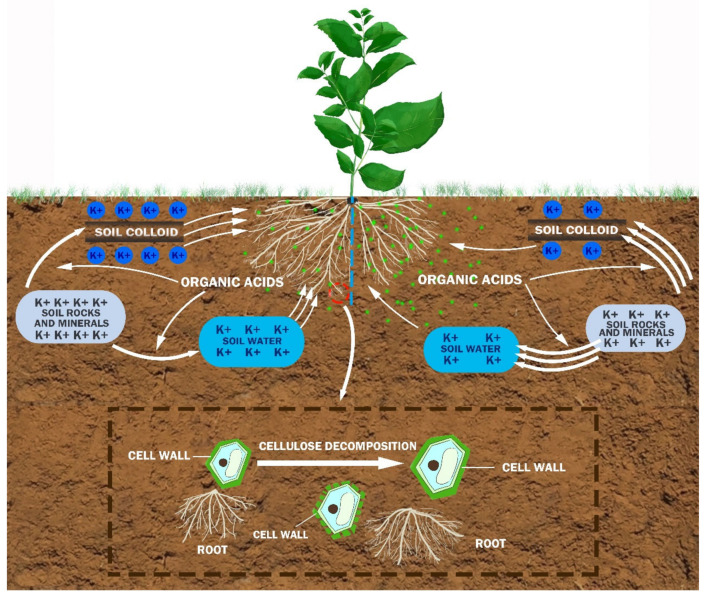
A conceptual diagram illustrating the foraging strategy of tea roots under K heterogeneity.

## Data Availability

Transcriptome sequencing data have been deposited in the National Center for Biotechnology Information (GSE198198). The other datasets generated during and/or analyzed during the current study are available from the corresponding author on reasonable request.

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
