# Peer review of "Root Foraging Strategy Improves the Adaptability of Tea Plants (Camellia sinensis L.) to Soil Potassium Heterogeneity"

_ijms, 2022, doi:10.3390/ijms23158585_

Round 1

Reviewer 1 Report

This manuscript on "Smart Root Foraging Strategy Improves the Adaptability of Tea (Camellia sinensis L.) Plants to Soil Potassium Heterogeneity" is novel and fits well under the scope of the journal.

I'd suggest the Authors to better specify in the introduction why they chose this specific breeding lines. Also the use of split-root rhizoboxes is appropriate but can raise questions on the validity of the study, since the roots were comprised in a narrow box filled with soils. The Authors should describe the system weighting on the pros and cons of their chosen experimental design.

Author Response

Response:

(1)Thank you for your advice. We have added the reason for choosing this specific breeding lines in the introduction as follows (lines 85-88): Tea plants are highly heterozygous because of their incompatibility and frequent interspecific hybridization. Taking this specific breeding lines as the research object makes this study based on the similar genetic background, which is more conducive to find out the target genes.

(2)Right. The system weighting on the pros and cons of the chosen experimental design has been added in the Methods as follows (lines 330-336): Split-root rhizobox is used to study the morphological, physiological and biochemical characteristics of roots. It can be used to study individual environmental factors (such as soil potassium). The plants growing in the split-root rhizobox are easy to master, and the growth conditions can be repeated. But there are also some limitations, which can raise questions on the validity of the study. Therefore, field experiments will be carried out later to further verify the wide application of this study. 

Reviewer 2 Report

The manuscript titled with" Smart Root Foraging Strategy Improves the Adaptability of Tea  (Camellia sinensis L.) Plants to Soil Potassium Heterogeneity". I see the manuscript is so important for the tea grower especially who they cultivated the tea in clay loam soil. But I have some comments on this work which could be summarized as follow;

The title should change into: Root Foraging Strategy Improves the Adaptability of Tea Plants (Camellia sinensis L.) to Soil Potassium Heterogeneity"

We deleted the word smart because, there is no mediator which regulates the release of K in the soil. Such as using the smart polymers which makes release adequate amount of K in the soil.

Abstract

Line 30; The present research provides a theoretical basis for a better understanding of the adaptation strategies of clonal woody plants to soil nutrient heterogeneity" should be changed into; The present research provides a practical basis for a better understanding of the adaptation strategies of clonal woody plants to soil nutrient availability.

Introduction

The authors are requested to write two more phrases on the economic importance of the tea plant in their country and all over the world. Also, the plant contents such as the antoxidant, dyes, anti bacterial, phenolic compounds etc and their importance for human health. This will show the importance of the plant that they work and the importance of this study.

At the end of the introduction  (line 87-95) authors did not mention the main aim of the study and its importance for the tea growers and producers.

Results

The authors did not mention the physiological parameters of the two tea plants types when both are stressed by K limitation. I think that they studied the effect only on the roots but they did not listed any results about the shoot system and plant content from polyphenols, antoxidant, etc, which will add a value of the stressed plants wither the tolerant for the K limitation or the sensitive one.  

Also, on the transcriptome level, which genes were linked to the K tolerance and its loci on which chromosome? The DNA marker which should be used in such studies to determine the key factor or the transfactor genes control the K tolerance.

Methods

Page 6 line 291

Determinations of Soil K Contents, Root morphology, and Plant K Contents should contains another test, soil microbiology, because there are many soil born bacteria can produce or absorb K from the soil  and or control the K availability for the plant. This mistake could be avoided if the authors make sterilization for the soil before the starting of the experiment.

Page 9

The cellulose decomposition genes were abundant on the K-enriched side may have promoted root proliferation for “1511”.

This statement should be confirmed by make a real time PCR for the RNA extracted from the two types of the tea plants under stress of the K deficiency compared with the control. This will confirm the relationship between the cellulose decomposing genes and the K tolerance.

Discussion

Discussion missed the part in which the authors should find a relationship between the physiological parameters and molecular finding associated with stressed plants compared with controls in the two examined tea plants.

Also, the role of the soil chemistry and the climate changes on the K availability for the tea plants in the absence of any help of soil microorganisms

Because the study was performed on strategic plants, a convey should be added in the end of the manuscript to help the tea growers.

Author Response

The manuscript titled with" Smart Root Foraging Strategy Improves the Adaptability of Tea (Camellia sinensis L.) Plants to Soil Potassium Heterogeneity". I see the manuscript is so important for the tea grower especially who they cultivated the tea in clay loam soil. But I have some comments on this work which could be summarized as follow;

The title should change into: Root Foraging Strategy Improves the Adaptability of Tea Plants (Camellia sinensis L.) to Soil Potassium Heterogeneity"

We deleted the word smart because, there is no mediator which regulates the release of K in the soil. Such as using the smart polymers which makes release adequate amount of K in the soil.

Response:

Thank you for your advice. We have changed the title into “Root Foraging Strategy Improves the Adaptability of Tea Plants (Camellia sinensis L.) to Soil Potassium Heterogeneity” (lines 1-2).

Abstract

Line 30; The present research provides a theoretical basis for a better understanding of the adaptation strategies of clonal woody plants to soil nutrient heterogeneity" should be changed into; The present research provides a practical basis for a better understanding of the adaptation strategies of clonal woody plants to soil nutrient availability.

Response:

Thank you for your advice. We have changed the previous sentence into “The present research provides a practical basis for a better understanding of the adaptation strategies of clonal woody plants to soil nutrient availability.” (lines 32-33)

Introduction

The authors are requested to write two more phrases on the economic importance of the tea plant in their country and all over the world. Also, the plant contents such as the antoxidant, dyes, anti bacterial, phenolic compounds etc and their importance for human health. This will show the importance of the plant that they work and the importance of this study.

At the end of the introduction (line 87-95) authors did not mention the main aim of the study and its importance for the tea growers and producers.

Response:

  • Thank you for your advice. The importance of the tea plant has been added in the Introduction as follows (lines 39-44):

Tea is, after water, the world’s most popular beverage. It contains abundant secondary metabolites like polyphenols, theanine, and caffeine, which offer a wealth of human health benefits. In the last decade, worldwide tea production has increased by∼66% in acreage, and has reached 5.3 million tons on 3.5 million hectares across 50 tea-growing countries (Food and Agriculture Organization of the United Nations statistics; www.fao.org/faostat/).

  • The main aim of the study and its importance for the tea growers and producers have been added in the Introduction as follows (lines 106-109):

Our findings will not only contribute to our further understanding of the adaptation strategies of tea plants to soil K availability, but also help to provides a reference for tea growers and producers to improve the potassium utilization by making full use of tea roots.

Results

The authors did not mention the physiological parameters of the two tea plants types when both are stressed by K limitation. I think that they studied the effect only on the roots but they did not listed any results about the shoot system and plant content from polyphenols, antoxidant, etc, which will add a value of the stressed plants wither the tolerant for the K limitation or the sensitive one.  

Also, on the transcriptome level, which genes were linked to the K tolerance and its loci on which chromosome? The DNA marker which should be used in such studies to determine the key factor or the transfactor genes control the K tolerance.

Response:

  • Thank you for your advice. The activities of antioxidant enzymes and polyphenol contents in shoot and root systems have been added in the Supplementary Figure S5. In both shoot and root, the activities of major antioxidant enzymes (SOD, POD and CAT) were significantly higher in “1511” than those in “1601” when both were stressed by K limitation. For shoot system, the content of procyanidins in “1511” was significantly higher than that in “1601” under K limitation, while there was no significant difference in total phenol content between the two tea plant types. For root system, the contents of both procyanidins and total phenol in “1511” were significantly higher than those in “1601” under K limitation. Therefore, the two genotypes had different tolerances for the K limitation. (lines 124-132)

The determination methods have also been added in the Methods as follows (lines 511-523): In order to analyze the physiological responses of the two tea plant types when both were stressed by K limitation, a culture experiment was carried out as described by our previous study [16]. First, tea plants were grown in normal nutrition solution for four weeks. Second, half of the above tea plants were transferred to a nutrient solution without K2SO4 (i.e., K starvation treatment) (SK), while the other half was transferred to a nutrient solution with K2SO4 as a control (i.e., control treatment) (CK). The tea plants were grown in normal and SK for eight weeks and the shoot and root were collected separately to measure the antioxidant enzyme activities and polyphenol contents. The activities of SOD, POD and CAT were determined according to the previous methods [58,59]. The procyanidin and total phenol contents were analyzed by the methods described by the previous research [60].

  • Through genome annotation and differential gene analysis, totally 1445 candidate genes were screened, which were linked to the K tolerance. Among these genes, there were 252 (17.44%) genes were newly discovered genes. Based on the comparison of reference genome, 1185 SNP loci were found in the candidate genes (82.01%). Among the 1185 SNP loci, 46.24% had an impact on the expression product, of which the variation of 9 loci (0.76%) had a high impact on the expression product, which will be focused on in the future. The relevant information of the potential molecular markers of K tolerance for tea plants has been showed in the Supplementary Table S6. (lines 177-184)

Methods

Page 6 line 291

Determinations of Soil K Contents, Root morphology, and Plant K Contents should contains another test, soil microbiology, because there are many soil born bacteria can produce or absorb K from the soil  and or control the K availability for the plant. This mistake could be avoided if the authors make sterilization for the soil before the starting of the experiment.

Page 9

The cellulose decomposition genes were abundant on the K-enriched side may have promoted root proliferation for “1511”.

This statement should be confirmed by make a real time PCR for the RNA extracted from the two types of the tea plants under stress of the K deficiency compared with the control. This will confirm the relationship between the cellulose decomposing genes and the K tolerance.

Response:

  • Soil microorganisms are indeed very important. Therefore, at the beginning of planting, the soil samples were sterilized by γ-irradiation (> 50 kGray) (Xiyue Radiation Technology Co., Ltd, NJ, China) (lines 322-323). However, because the planting time is so long (one year), some microorganisms will still breed in the process of cultivation Most of these newly bred microorganisms are closely related to tea roots. This paper focuses on the root foraging strategy, so the soil microbiology is not considered in this paper, but further research will be carried out in the future.
  • Thank you for your advice. In order to confirm this statement, we conducted a real time PCR on three genes involved in cellulose decomposition, and found that the results of the real time PCR were significantly positively correlated with the total root length (Supplementary Figure S4). However, K tolerance is interfered by more factors (such as antioxidant enzyme activity, etc.), so we do not focus on the relationship between cellulose decomposition genes and potassium tolerance. (lines 267-270)

Discussion

Discussion missed the part in which the authors should find a relationship between the physiological parameters and molecular finding associated with stressed plants compared with controls in the two examined tea plants.

Also, the role of the soil chemistry and the climate changes on the K availability for the tea plants in the absence of any help of soil microorganisms

Because the study was performed on strategic plants, a convey should be added in the end of the manuscript to help the tea growers.

Response:

  • Thank you for your advice. A relationship between the physiological parameters and molecular finding associated with stressed plants compared with controls in the two examined tea plants has been added in the Discussion as follows (lines 278-290): K transporters and channels are critical in the K+ absorption and translocation in plants [30]. In our study, a K transport related gene (TEA016242) was significantly up-regulated in “1511”, while there was no K transport related gene up-regulated in “1601” under K stress. Meanwhile, the leaf and root K concentrations were significantly higher in “1511” than those in “1601”. Therefore, for K absorption, the molecular and physiological results were consistent under K stress. When plants are stressed by K deficiency, ROS will be produced in large quantities. SOD is the first defense line to remove ROS in plant cells [44]. In this study, a SOD gene (TEA008360) was significantly up-regulated only in “1511” under K stress. For “1601”, no genes related to antioxidation were found under K stress. Meanwhile, the activity of SOD was significantly higher in “1511” than that in “1601” when both roots were stressed by K limitation. Therefore, for antioxidation process, the molecular and physiological results were also consistent under K stress.
  • Thank you for your advice. A convey should be added in the end of the manuscript to help the tea growers as follows (lines 300-305): In the absence of any help of soil microorganisms, soil chemistry and climate change will also have significant impacts on the K availability for the tea plants, while this study mainly focused on the role of strategic plants in improving soil K utilization. Therefore, tea farmers should also fully consider the impact of soil chemistry and climate change on soil nutrient availability.

Round 2

Reviewer 1 Report

The authors addressed all my comments and the paper is now ready for publication.